# Host influence on the eukaryotic virome of sympatric mosquitoes and abundance of diverse viruses with a broad host range

**Côme Morel[1], Patricia Gil[1], Antoni Exbrayat[1], Etienne Loire[1], Florian Charriat[1], Baptiste Prepoint[1], Celine Condachou[1], Geoffrey Gimonneau[1,2], Assane Gueye Fall[2], Biram Biteye****[2], Momar Talla Seck[2], Marc Eloit[3,4,5], Serafin Gutierrez**[1] *

**1** ASTRE, Cirad, INRAe, Univ Montpellier, Montpellier, France, **2** Institut Sénégalais de Recherches Agricoles/Laboratoire National de l'Elevage et de Recherches Vétérinaires (ISRA), Dakar-Hann, Senegal, **3** Institut Pasteur, Université Paris Cité, Pathogen Discovery Laboratory, Paris, France, **4** Institut Pasteur, Université Paris Cité, The WOAH (OIE) Collaborating Center for The Detection and Identification in Humans of Emerging Animal Pathogens, Paris, France, **5** Ecole Nationale Vétérinaire d'Alfort, University of Paris-Est, Maisons-Alfort, France

* serafin.gutierrez@cirad.fr

**Data Availability Statement:** All raw reads and contigs files are available from the GenBank database (BioProject PRJNA814001).

## Abstract

Mosquitoes harbor a large diversity of eukaryotic viruses. Those viromes probably influence mosquito physiology and the transmission of human pathogens. Nevertheless, their ecology remains largely unstudied. Here, we address two key questions in virome ecology. First, we assessed the influence of mosquito species on virome taxonomic diversity and relative abundance. Contrary to most previous studies, the potential effect of the habitat was explicitly included. Thousands of individuals of *Culex poicilipes* and *Culex tritaeniorhynchus*, two vectors of viral diseases, were concomitantly sampled in three habitats over two years. A total of 95 viral taxa from 25 families were identified with meta-transcriptomics, with 75% of taxa shared by both mosquitoes. Viromes significantly differed by mosquito species but not by habitat. Differences were largely due to changes in relative abundance of shared taxa. Then, we studied the diversity of viruses with a broad host range. We searched for viral taxa shared by the two *Culex* species and *Aedes vexans*, another disease vector, present in one of the habitats. Twenty-six out of the 163 viral taxa were found in the three mosquitoes. These taxa encompassed 14 families. A database analysis supported broad host ranges for many of those viruses, as well as a widespread geographical distribution. Thus, the viromes of mosquitoes from the same genera mainly differed in the relative abundance of shared taxa, whereas differences in viral diversity dominated between mosquito genera. Whether this new model of virome diversity and structure applies to other mosquito communities remains to be determined.

**Funding:** This work was funded by the European Union's Seventh Framework Programme through grant Vmerge (FP7-613996).

**Competing interests:** The authors have declared that no competing interests exist.

## Introduction

Certain mosquito species transmit pathogens that impose an enormous burden on human health worldwide. Climate change, globalization and urbanization could further increase this burden through, among others, a larger geographical distribution of mosquito vectors [1]. However, mosquito control is challenging mainly due to insecticide resistance [2]. As a result, an intense research effort is being conducted to find new strategies to limit pathogen transmission by mosquitoes [2].

The study of the viral community, or virome, infecting mosquitoes is a recent research field that could lead to the development of new control tools against pathogen transmission. Recent advances in metagenomics have unveiled that mosquitoes harbor a large and diverse virome that, as usually observed in terrestrial eukaryotes, is mainly formed by RNA viruses [3]. Contrary to the well-known mosquito-borne viruses of humans, also known as arboviruses, most mosquito viruses do not seem to infect vertebrates. Little is yet known on their ecology and they are often thought to be mosquito commensals that are vertically transmitted [4]. Interestingly, a growing body of work suggests that they can influence different aspects of mosquito physiology and the transmission of pathogens [5,6]. For example, mosquito-specific viruses from different families can modulate the mosquito infection rate of arboviruses [7–9].

Beyond the influence of a single mosquito-specific virus, arbovirus transmission could be influenced by interactions between an arbovirus and the virome as a whole. This potential influence of the whole virome is predicted by theory [10]. For example, the higher the species richness in the virome, the more likely that niche overlap, and thus competition, takes place between an invasive arbovirus and another member of the virome [11]. Studies on such arbovirus-virome interactions are almost nonexistent [12]. This paucity of data is mainly due to our limited understanding of the diversity and abundance patterns in mosquito viromes, a situation shared by the viromes of eukaryotes in terrestrial ecosystems in general [13,14]. Detailed characterizations of such patterns are thus required to design hypothesis-driven studies of virome influence on arbovirus transmission [12,15].

To date, most studies on the virome ecology in mosquitoes have focused on the influence of the host species, a key question in microbial community ecology. Most studies have compared virus diversity and a few studies have also compared virome structure (*i.e.* relative abundance) between mosquito species [15–19]. Overall, differences in both diversity and structure have been observed, suggesting that mosquito taxonomy shapes the eukaryotic virome. These observations follow a widespread prediction on virome diversity and structure based on the high specificity of the molecular interactions between viruses and host factors [16]. That is, the viral diversity able to infect a given mosquito species would be mainly restricted to those viruses that have co-evolved with their host. A vertical transmission, the transmission mode usually hypothesized for most mosquito-specific viruses [4], should provide the conditions for tight co-evolution and, thus, virome divergence between different mosquito species along their evolution. However, methodological biases in most studies preclude a robust analysis of the influence of the host species, leaving the question largely unanswered.

The main methodological bias is that mosquito species are rarely simultaneously collected in different sites or habitats [16–18,20,21]. This situation is surprising because the habitat could have a major influence on the degree of virome similarity between sympatric mosquito species. More precisely, the habitat could define the probability of horizontal transmission of viruses between mosquito species. Although different mosquito species tend to exploit different niches in a given habitat, a certain degree of niche overlap is frequently observed, especially in habitats with a limited niche diversity. Niche overlap can thus allow for virus exchanges between mosquito species. For example, virus transmission between different species can take

place in the larval breeding sites, in which larvae of different species are present simulta-
neously. Thus, the large diversity of mosquito ecologies and habitats leads to a complex situa-
tion in which virome diversity and structure in sympatric mosquito species could be defined
by the mosquito species, the habitat or both. Hence, the comparison of the virome between
sympatric mosquito species in different habitats seems paramount to robustly infer the relative
influence of host species over that of the collection site. Moreover, virome comparisons in hab-
itats with intense niche overlap can allow to determine the taxonomic diversity and relative
abundance of viruses with a wide host range in eukaryotic viromes, a fascinating but largely
unexplored question.

Here, we address two questions on the eukaryotic virome of mosquitoes. The first question
is the influence of the mosquito species on the taxonomic diversity of viruses and its structure.
The second question is the diversity and prevalence of generalist viruses (*i.e.* viruses with a
broad host range). To explore the first question, the eukaryotic viromes of two important vec-
tors of arboviruses, *Culex poicilipes* and *Culex tritaeniorhynchus*, were compared. These mos-
quitoes differ in certain aspects of their ecologies. For example, they differ in the choice of
breeding sites depending on water pollution, with *Culex poicilipes* being less refractory to pol-
luted waters [22]. Nevertheless, they can be found together in the same breeding sites in certain
habitats (e.g. here observed in the temporary ponds in the Ferlo; A.G. Fall, personal communi-
cation). Sympatric mosquito populations were simultaneously collected in three habitats with
different water ecosystems, and thus mosquito breeding sites, situated hundreds of kilometers
apart. Contrary to most previous studies, sampling involved two years and thousands of indi-
viduals per mosquito species. Meta-transcriptomics analysis of viromes unveiled that, despite a
large overlap in virus diversity between mosquitoes, host species mainly defined virome struc-
ture. Thus, differences in virome structure were mostly driven by changes in the relative abun-
dance of shared viral families. The large proportion of shared viruses led us to study the
diversity and prevalence of generalist viruses. To do so, we compared the viromes of the *Culex*
species with that of a mosquito species from another genus and with a distinct ecology, *Aedes
vexans*. The comparison was done in the arid habitat with a limited diversity of mosquito
breeding sites. This environment should favor niche overlap between mosquito species and, in
turn, the interspecies transmission of generalist viruses. Despite important differences in virus
diversity between the two mosquito genera, viruses with multi-genus host ranges were rela-
tively abundant and diverse. A search in public databases showed that several generalist viruses
were not only present in other mosquito species but also had a geographical distribution
encompassing several continents. Overall, our results suggest that differences in virome diver-
sity and structure can change along with taxonomic distance in the studied mosquito species
and sites. Increasing taxonomic distances would lead from changes in relative abundance of
shared viruses to changes in taxonomic diversity. Moreover, our work calls for studies on the
prevalence, origin and role of generalist viruses in mosquito viromes.

## Materials and methods

### Study sites

Sampling sites have been previously described [22]. Mosquito sampling was carried out in
three localities in Northern Senegal (Diama, Keur Momar Sarr and Younoufere), between 57–
234 km distant from each other (Fig 1). The climate is similar in all sites. This semi-arid cli-
mate is characterized by a rainfall between 100 and 500 mm/year during the 3-month rainy
season and a long dry season. Diama (16˚12′41.4′′N, 16˚23′31.6′′W) is a village on the bank of
the Senegal river, a large perennial stream (mean water discharge = 640 m$^3$/s). Keur Momar
Sarr (KMS) (15˚56′51.7″N, 15˚56′22.2″W) is a town on the bank of Lake Guiers, a large fresh

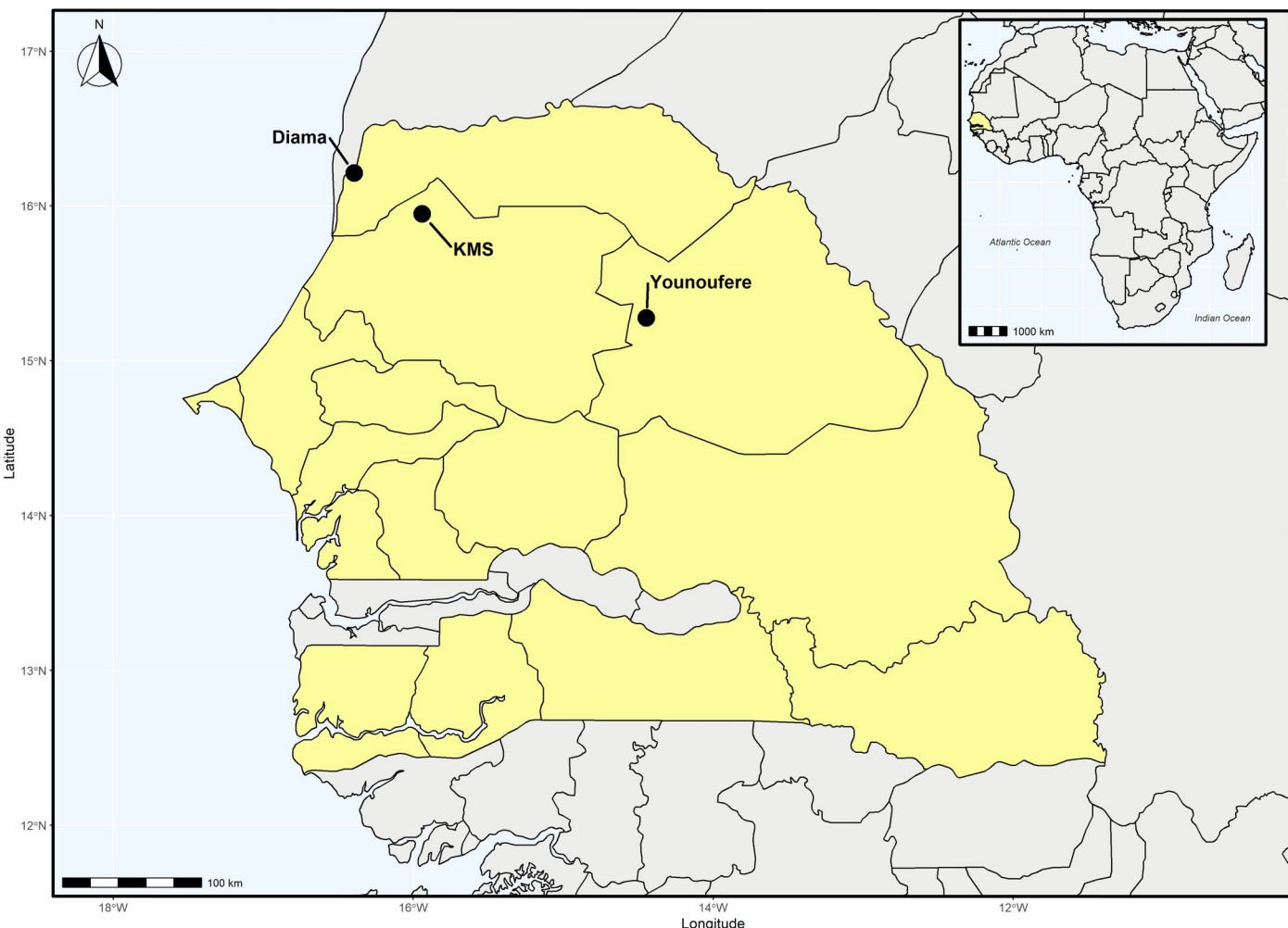

**Fig 1. Location of the three sampling sites in northern Senegal.** KMS: Keur Momar Sarr. Upper right corner: Map of Africa with Senegal coloured in yellow. Main figure: The solid circles correspond to the sampling points: Diama (River), Keur Momar Sarr (Lake) and Younoufere (Ponds).

water body enduring drastic changes in water level between seasons (seasonal drop in surface from 300 to 170 km$^2$). Younoufere village (15˚16′08.7″N, 14˚27′52.5″W) is located in the Ferlo region, an arid steppe. In this last site, mosquito breeding mainly takes place in scattered ponds that usually hold water only during the rainy season. Each site has specific mosquito communities [22] and, thus, potentially communities of mosquito-associated viruses. For example, *C. tritaeniorhynchus* was the most abundant species (around 50%) in collections from the sites with permanent watercourses (Diama and Keur Momar Sarr) the two sampling years, whereas *A. vexans* was the most abundant species (95%) in the site with temporary watercourses (Younoufere) [22]. To facilitate habitat identification, sites are named in the text after the type of surface water in each site: "River" for Diama, "Lake" for Keur Momar Sarr and "Ponds" for Younoufere.

## Mosquito collection and identification

Mosquito collection has been previously described in detail [22]. Briefly, sampling was carried out in each site once per month during the main mosquito season, from July to November, in

2014 and 2015. Mosquitoes were trapped during two consecutive nights using $CO_2$-baited CDC light traps (BioQuip). Mosquitoes were processed under temperature conditions minimizing RNA degradation. These conditions included freezing in dry ice right after collection on the field, species sorting on a chill table, and storage of females at -80˚C. A total of 2 029 females of *Culex poicilipes* (645 for the River, 1 169 for the Lake, 215 for the Ponds) and 15 180 females of *Culex tritaeniorhynchus* (7 750 for the River, 7 247 for the Lake, 183 for the Ponds) were collected (Table 1). A total of 23 037 females of *Aedes vexans* were collected in the environment with temporary ponds, the only site in which this species was abundant (Table 1). Transfer of mosquitoes to the ASTRE laboratory was done under import permit 2016018 granted by the French Ministry of Food and Agriculture.

## Library preparation and sequencing

Library preparation and sequencing was carried out as described [23]. Pools of around 30 non-engorged females from the same species, site and date were used for total RNA isolation with the Nucleospin RNA virus kit (Macherey Nagel). Twenty-two RNA libraries (six libraries for each *Culex* species and ten libraries for *A. vexans*) were generated by pooling RNA extractions from the same site and date. On average, 1 830 mosquitoes were used per library (min./ max. = 85 / 4 062, Table 1). Libraries were retro-transcribed to cDNA (SuperScript IV reverse

**Table 1. Metadata and sequencing output of the 22 libraries of this study.**

| Sample | Species | Site | Type | Year | Mosquito number | Raw reads x10$^7$ | Viral reads x10$^4$ | Filtered viral reads x10$^4$ |
|---|---|---|---|---|---|---|---|---|
| PF4 | *Culex poicilipes* | Younoufere | Ponds | 2014 | 112 | 8.18 | 8.04 | 7.93 |
| PF5 | *Culex poicilipes* | Younoufere | Ponds | 2015 | 103 | 7.55 | 1.82 | 1.78 |
| PD4 | *Culex poicilipes* | Diama | River | 2014 | 94 | 6.82 | 2.32 | 2.27 |
| PD5 | *Culex poicilipes* | Diama | River | 2015 | 551 | 6.1 | 7.62 | 7.5 |
| PK4 | *Culex poicilipes* | Keur Momar Sarr | Lake | 2014 | 786 | 8 | 11.87 | 11.79 |
| PK5 | *Culex poicilipes* | Keur Momar Sarr | Lake | 2015 | 383 | 10.44 | 7.52 | 7.42 |
| TF4 | *Culex tritaeniorhynchus* | Younoufere | Ponds | 2014 | 98 | 6.04 | 1.17 | 1.14 |
| TF5 | *Culex tritaeniorhynchus* | Younoufere | Ponds | 2015 | 85 | 6.79 | 2.34 | 2.32 |
| TD4 | *Culex tritaeniorhynchus* | Diama | River | 2014 | 3 829 | 8.46 | 10.6 | 10.48 |
| TD5 | *Culex tritaeniorhynchus* | Diama | River | 2015 | 3 921 | 7.75 | 5.56 | 5.47 |
| TK4 | *Culex tritaeniorhynchus* | Keur Momar Sarr | Lake | 2014 | 4 062 | 7.43 | 1.82 | 1.77 |
| TK5 | *Culex tritaeniorhynchus* | Keur Momar Sarr | Lake | 2015 | 3 185 | 7.5 | 3.61 | 3.53 |
| **Total** | | | | | **17 209** | **91.06** | **64.3** | **63.39** |
| DIA41 | *Aedes vexans* | Younoufere | Ponds | 2014 | 8 640 | 6.48 | 10.13 | 10.09 |
| DJI41 | *Aedes vexans* | Younoufere | Ponds | 2014 | 7 710 | 6.45 | 10.69 | 10.57 |
| NAC41 | *Aedes vexans* | Younoufere | Ponds | 2014 | 1 590 | 7.25 | 5.56 | 5.52 |
| DIA42 | *Aedes vexans* | Younoufere | Ponds | 2014 | 690 | 8.61 | 13.25 | 13.21 |
| DIA43 | *Aedes vexans* | Younoufere | Ponds | 2014 | 1 320 | 6.63 | 10.42 | 10.36 |
| DJI51 | *Aedes vexans* | Younoufere | Ponds | 2015 | 132 | 7.46 | 7.95 | 7.92 |
| DIA52 | *Aedes vexans* | Younoufere | Ponds | 2015 | 1 579 | 6.81 | 21.52 | 21.06 |
| DJI52 | *Aedes vexans* | Younoufere | Ponds | 2015 | 1 041 | 7.51 | 22.17 | 21.99 |
| NAC52 | *Aedes vexans* | Younoufere | Ponds | 2015 | 196 | 6.8 | 11.61 | 11.57 |
| DIA53 | *Aedes vexans* | Younoufere | Ponds | 2015 | 139 | 6.09 | 16.01 | 15.61 |
| **Total** | | | | | **23 037** | **70.09** | **129.32** | **127.9** |

The variable "Filtered viral reads" stands for the number of reads after the quality filtering of viral taxonomic units (VTU) (for a definition of VTU see Material and Methods).

transcriptase, Invitrogen) with random hexamers, and the cDNA was amplified with phi29 polymerase and random hexamers [24]. Libraries were prepared using the TruSeq DNA Nano Library Prep kit (Illumina) and sequenced using an Illumina HiSeq2000 sequencer (single-end 150 base pair format) to an expected depth of approximately 80 million reads. Library preparation and sequencing were performed by DNAVision (Charleroi, Belgium).

## Bioinformatic and statistical analyses

The bioinformatic analysis of sequences was done with the SnakeVir pipeline as previously described [23]. Adapters and poor-quality sequences were removed from Illumina reads with Cutadapt version 1.6 [25]. The resulting reads were removed if they mapped with BWA version 0.7.15 [26] on rRNA sequences belonging to bacterial or dipteran sequences (SILVA rRNA bases: SSURefNr99 and LSURef, 18/01/2017; dipteran base: release 138) [27]. Then, a two-round *de-novo* assembly was performed using Megahit 1.1.2 [28] and CAP3 [29]. Reads from all the libraries of the same mosquito genus were pooled and used as input in the *de-novo* assembly to generate a non-redundant set of contigs. A homology search was done with the resulting contigs and DIAMOND [30] against the NCBI nr database (publication date: 05/09/ 2021) (e-value threshold = $10^{-3}$; minimum alignement length = 75 bp). Quantification of the number of reads mapping per contig and library was carried out with BWA 0.7.15 [26]. Contigs with a virus as best hit (*i.e.* the accession with the lowest e-value found by Diamond) were screened for a potential origin from endogenous viral sequences with a Blastn against the NCBI nt database. Contigs with a non-viral best hit with a coverage above 25% were considered as potential EVEs and removed [31].

We generated an arbitrary taxon, similar to the species level, into which all the contigs with best hits associated with the same virus species were grouped. This taxon cannot be always considered as a virus species because contigs from new and closely-related viruses can be associated to the same known virus and, thus, to the same taxon. Moreover, contigs with different genes of the same genome can be clustered in different taxa if their best-hit species differs (e.g., the capsid and RdRP genes of the same new virus could have different best-hit species). This clustering method has often been used due to the lack of clustering methods for contigs generated with shotgun sequencing from eukaryotic hosts [12,19,20,31–34]. There is not a clear identification of the species-like taxon generated with this method in the literature. Thus, we named this taxon as "viral taxonomic unit" (VTU) to distinguish it from "operational taxonomic unit" (OTU) or "viral taxonomic operational unit" (vOTU), both terms with specific definitions that differ from that of a VTU [35,36]. For example, the definition of OTU usually involves a cutoff based on the percent similarity of overlapping sequences (*e.g.*, 97% similarity). In shotgun sequencing, we cannot use a cutoff based on the alignment of contigs of a VTU due to their non-overlapping nature (i.e., they are distributed over different regions of the genome). VTU names were formed by combining the name of the best-hit species with the average percent identity at the amino acid level of the contigs associated with this best hit. For this purpose, and to facilitate analysis with R software, the percent identity was placed at the end of the best hit name after a double underscore. Thus, and contrary to previous studies using the same clustering, the inclusion of the mean percent identity allows to determine whether the VTU contains sequences from a new virus species. Contigs were considered to belong to a known virus if their percent identities at amino acid level with sequences from the known virus species were above 90%. This cutoff was based on the threshold usually proposed by ICTV groups for species demarcation.

The average contig length per VTU was 3 300 bp (max/min: 20 251/404 bp, see S2 and S3 Tables). To validate VTU detection in different mosquito species, contigs of each shared VTU

were compared between mosquito species with Blastn. A VTU was considered as present in several mosquito species if any of its contigs in a given mosquito had percent identity above 80% at the nucleotide level with those found in another mosquito species. Taxonomic affiliation of VTUs to families was often hampered by a lack of a family rank in the taxonomy of best-hit species. Thus, VTUs were grouped into a family-like level called "cluster" as previously described [31]. Briefly, the cluster was defined as the likely family of the best-hit species based either on GenBank information or on the taxonomy of the closest relatives of the best-hit species found with Blastn searches. The statistical analyses were done at the cluster level to limit the influence of the potential biases in VTU clustering mentioned above.

Several steps were carried out to validate VTU detection. The first step was applied to each library separately and consisted in setting the read number to zero for all VTUs with less than 10 reads in a given library. That is, those VTUs were considered undetected in the library. This threshold was based on previous comparisons of PCR and metagenomics for virus detection in mosquito samples [23,31,37]. After this step, a VTU was considered detected if the sum of its reads in all libraries was at least 100 reads. The 100-read threshold was based on the distribution of reads per VTU. That is, the cut-off was set at the value when the distribution approached an asymptote. Any VTU with less than 100 reads in total was removed from the dataset. A final step was carried out to limit the influence of a potential cross-contamination between libraries. This analysis focused on the VTUs with the highest number of reads, that is those VTUs that were more likely to lead to cross-contamination. For the most abundant VTUs (i.e. with a total number of reads greater to the third quartile of the distribution), we set a minimal read number per library to be considered as detected. This threshold was set at 5% of the reads in the library with the highest number of reads. Read counts were set to zero in a given library if below the threshold (i.e., the VTU was considered not detected). The threshold can be considered as conservative as it reached over 2000 reads for certain VTUs. After VTU detection filters, we removed VTUs corresponding to best hits found in hosts other than Arthropoda in order to eliminate any viral sequence that may not derive from mosquito viruses.

All statistical analyses were performed on Rstudio (v.4.1.1). The Venn diagrams were generated with the *ggvenn* package [38]. The pie charts were created with the *webr* package [39]. Heatmaps were generated with the *superheat* [40] and the *ComplexHeatmap* [41] packages. Measures of alpha and beta diversity were done with the *vegan* package [42] from read counts normalized using Total Sum Normalization [43]. The maps were created with *rnaturalearth* [44] and *rgdal* [45] packages and the shapefile of the Senegal map was uploaded from gadm.org website. Finally, the other figures were made using *ggplot2* [46] and *phyloseq* [47] packages.

## Analysis of the host and geographical ranges of selected viruses

A Blastn search against the Genbank database was performed to determine if viruses in our dataset had been previously found associated with other mosquito species and geographic areas. We considered that a VTU included a virus that had been previously detected if it included contigs with more than 80% identity at the nucleotide level with sequences from its best hit species, as well as a minimum alignment coverage of 90%. An analysis of the bibliography on the selected viruses was carried out in PubMed (date 04/2022) to determine their host and geographical ranges (S1 Table).

## Inclusivity in global research

Additional information regarding the ethical, cultural, and scientific considerations specific to inclusivity in global research is included in the (S1 Checklist).

## Results

### A large taxonomic diversity in the viromes of two sympatric mosquitoes

We compared the virome of two mosquito species, *C. poicilipes* and *C. tritaeniorhynchus*, simultaneously collected in three distant habitats over two years. For this purpose, a large number of mosquitoes (17 209 individuals) was analyzed to maximize the chances of an exhaustive exploration of virome diversity. Twelve libraries were generated from mosquito collections and analyzed with metatranscriptomics [23]. Each library included all the adult females of a given mosquito species captured in a habitat over a year (1 400 individuals per library on average, Table 1). The variable "year" was considered as a proxy of replicates for each host/site combination and was included in statistical analyses either as a random effect or to assess replicate influence.

Sequencing yielded 910 million reads. A bioinformatic analysis, based on a mapping of all reads on viral contigs, identified 642 000 virus-like reads (0.07% of the total reads) and 110 VTUs (Table 1). The output in virus-like reads significantly varied among libraries, up to a ten-fold (Exact Wilcoxon Signed-Rank Test, p-value < 0.001). No significant effect of mosquito number on the number of either virus-like reads or VTUs was detected (Spearman rho > 0.31, p-values > 0.1). Rarefaction curves further supported that sampling effort in terms of virus-like reads was satisfactory, including for libraries with the lower mosquito numbers (S1A Fig). Most VTUs were related to RNA viruses associated with arthropod hosts as usually found in mosquito viromes (91% of the VTUs) [48].

Several filtering steps led to a final dataset with 95 VTUs, representing 96% of the virus-like reads. The taxonomic diversity was relatively large, including 25 family-like clusters, 16 orders and all the five phyla of RNA viruses (Figs 2 and S2A). Around three quarters of the VTUs had percent identities at the amino-acid level below 90% with the virus species of their best hit. This observation strongly suggested the discovery of new virus species as usually observed in mosquito viromes (S3A Fig). The read distribution of VTUs and clusters was highly skewed within each library, with most reads provided by a few taxa, as previously observed [16,49,50] (Figs 2 and S4). For example, four out of the 25 clusters represented more than 50% of the virus-like reads (Fig 2). This observation was found in the viromes of the two *Culex* species, although with different dominant taxa (Figs 2 and S4).

### Viromes differed depending on the host despite a large diversity overlap

The overlap in virus diversity was large between mosquito species, habitats and years (S5 Fig). Shared taxa between the two mosquitoes represented 92% of the clusters and 75% of the VTUs (S5 Fig). We then analyzed the potential influence of host and habitat on virome diversity and abundance both at the cluster and the VTU level. For the sake of conciseness, only the results from the analysis of the cluster dataset are presented in the main text. The results for the VTUs followed those obtained with the clusters (available in the Supplementary Material, S6 Fig). As expected from the high diversity overlap, taxon richness was not significantly different for host and habitat (Mann Whitney Wilcoxon tests for host, p-value = 0.80; Kruskal Wallis test for habitat, p-value = 0.183) (Fig 3A). Shannon and Simpson indices were significantly different between mosquito species but not between habitats (Mann Whitney Wilcoxon tests for host, p-value = 0.041; Kruskal Wallis test for habitat, p-value = 0.874) (Fig 3A). Viromes tended to group depending on host, although not perfectly, when analyzed using non-parametric multi-dimensional scaling and Bray-Curtis dissimilarities or hierarchical clustering (Figs 2 and 3B). Virome clustering depending on host was supported in a PERMANOVA (p-value = 0.011). The latter analysis did not detect a significant effect of habitat, year or a host-habitat

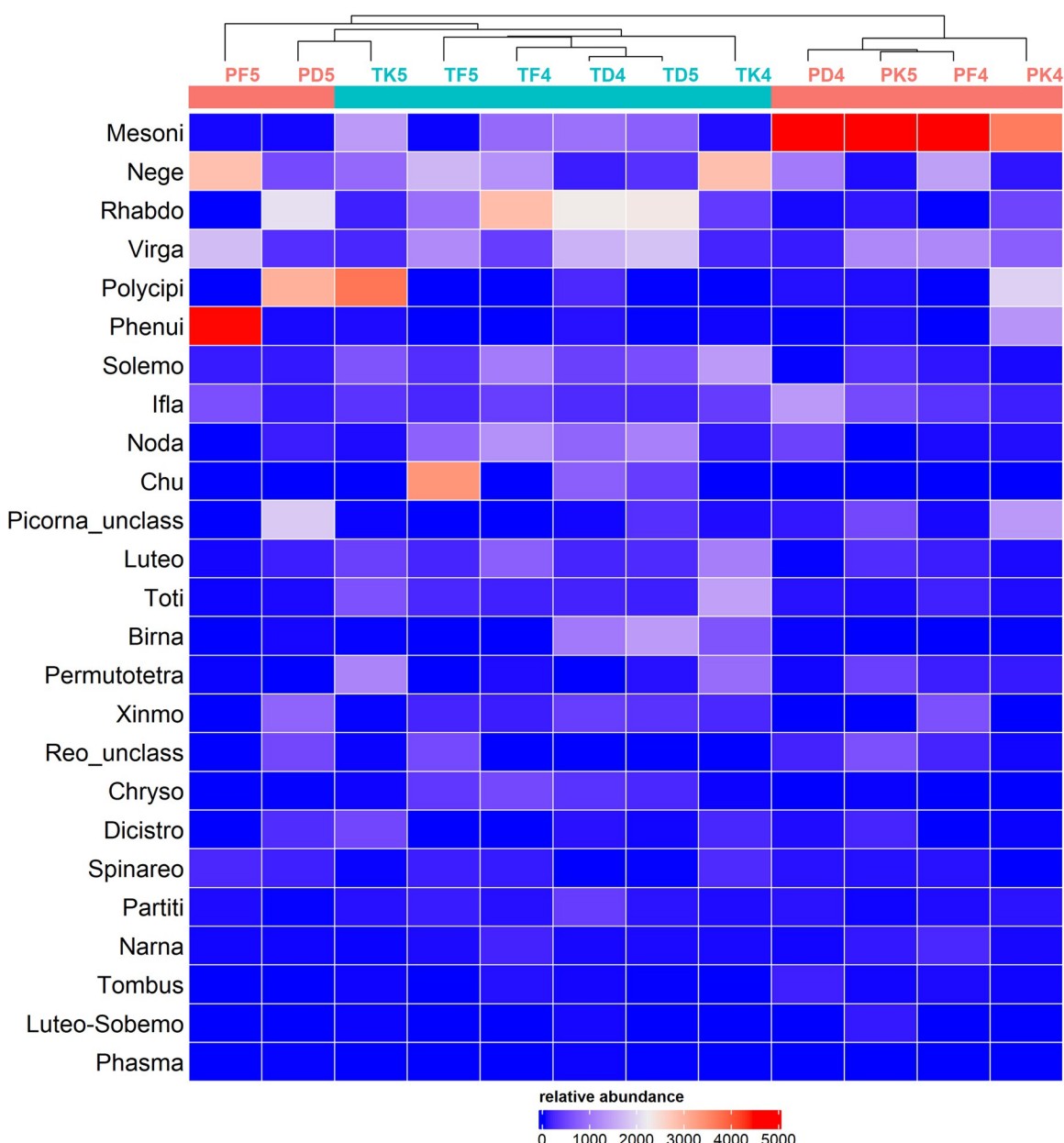

**Fig 2. Distribution and relative abundance of viral clusters in the libraries of *Culex poicilipes* and *Culex tritaeniorhynchus*.** Library names are indicated on top of the heatmap (see Table 1 for explanation of acronyms), along with a hierarchical clustering, and library colour indicates mosquito species (red: *Culex poicilipes*, blue: *Culex tritaeniorhynchus*). Tile colour stands for read abundance; the more abundant a cluster, the warmer the colour. The clusters are ranked following total read abundance.

interaction (p-values > 0.29). Overall, our results showed a significant influence of host but not habitat on the viromes of the two *Culex* mosquitoes.

We analyzed the nature of the differences in virome diversity and structure between the *Culex* mosquitoes. More specifically, a presence/absence dissimilarity index (Sorensen dissimilarity) was estimated between pairs of samples, each sample from a different mosquito species, to determine the extent of the overlap in the taxonomic diversity of family-like clusters. Sorensen's dissimilarities were often below 0.5 (mean Sorensen's dissimilarity = 0.15), a situation

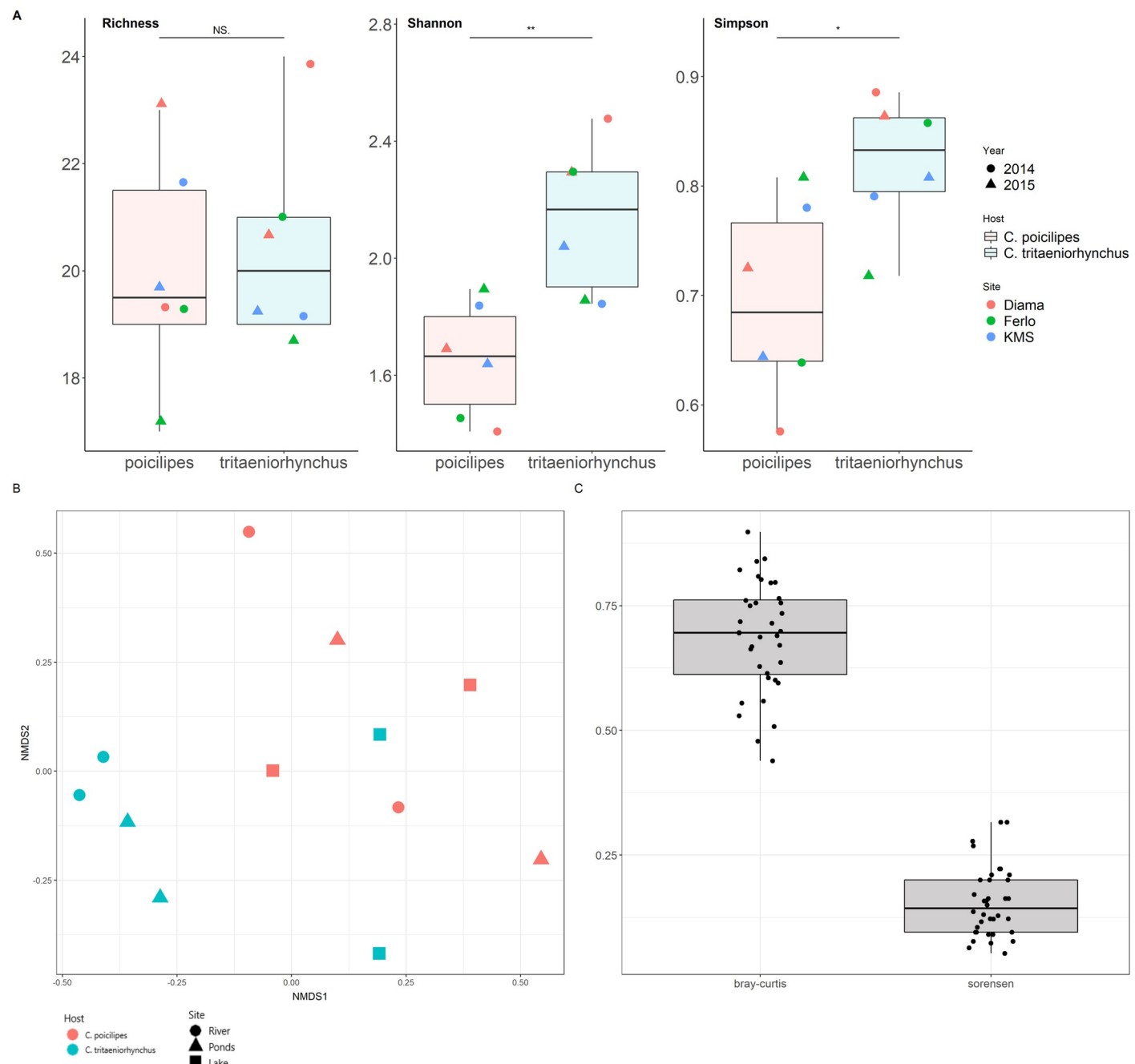

**Fig 3. Analysis of alpha and beta diversities in the viromes of two *Culex* mosquitoes. A.** Distribution of species richness, Shannon and Simpson indices estimated from cluster data among libraries of *Culex poicilipes* (in red) and *Culex tritaeniorhynchus* (in blue). Dot color indicates the habitat while dot shape represents year. The significance of the comparison between distributions in each of the mosquito species is shown above boxplots (Wilcoxon Mann-Whitney test; NS: Non-significant, *: p-value < 0.05, **: p-value < 0.01). **B.** Non-metric multidimensional scale analysis with Bray-Curtis dissimilarities obtained from the cluster data of the two *Culex* species. Dot color indicates mosquito species and dot shape represents habitat. **C.** Distributions of Sorensen (for presence-absence data) and Bray-Curtis (for abundance data) dissimilarities between viromes of the two *Culex* mosquitoes. Each point represents a dissimilarity index value between two libraries belonging each to a different mosquito species.

showing a large overlap in cluster diversity. Then, we estimated another dissimilarity index, the Bray-Curtis dissimilarity, for the same sample pairs. The Bray-Curtis dissimilarity allows

to explore differences in relative abundance of taxa between two samples. The Bray-Curtis dissimilarities were usually above 0.5 (mean Bray-Curtis dissimilarity = 0.69) (Fig 3C). These observations suggested that differences in relative abundance of shared clusters dominated over differences in taxonomic diversity. The analysis of the VTU dataset suggested a similar pattern, although differences in diversity were more important (means of Sorensen and Bray-Curtis dissimilarities = 0.49 and 0. 86, respectively; S6 Fig).

## Viruses with broad host ranges can be an important and diverse fraction of mosquito viromes

The above results suggested that many VTUs had a host range including two mosquito species of the *Culex* genus. To further explore their host range width, we extended the virome comparison to an additional mosquito species from another genus, *Aedes vexans*. This mosquito was the most abundant species in the Ponds but it was not collected in sufficient numbers in the other sites. The comparison of virome diversity was thus possible in the habitat in which superficial water is scarce and sharing of larval breeding sites takes place between the *Culex* and the *Aedes* species. Abundant catches of *A. vexans* allowed the analysis of a number of individuals and libraries in the same range as that of all the *Culex* species together (2300 females per library on average, ten libraries, Table 1). Again, we observed large variations in virus-like reads between libraries but variation was not explained by mosquito number (Spearman rho = -0.26, p-value = 0.47). Rarefaction curves suggested adequate sampling effort in terms of virus-like reads (S1B Fig).

After VTU filtering, the *A. vexans* dataset comprised 96 VTUs, all related to arthropod viruses. Taxonomic diversity was again large, involving 32 clusters and 20 orders (S2B Fig). Moreover, a fraction of the VTUs included novel virus species (S3B Fig).

Our goal was to identify viruses with a wide host range and thus we focused our analysis on the overlap in VTU diversity. In contrast to the situation between the two *Culex* species, viromes clearly differed in taxonomic diversity between the *Culex* and *Aedes* species, both at the VTU and cluster levels (Figs 4A and S7). The three mosquitoes shared 26 out of the 163 VTUs (16%) (Fig 4B). Those 26 VTUs encompassed a large taxonomic diversity including 14 clusters, 10 orders and the five phyla of true RNA viruses (Fig 4C). Ten shared VTUs probably included new virus species. Most of the shared VTUs (24 out of 26) were found in all sites at least one year, supporting their presence over a transect of 230 km in Northern Senegal (S8 Fig).

We explored the host range and geographical distribution of 16 known viruses found among the shared VTUs. To this end, we scanned GenBank and literature for records of those viruses. Host ranges of the selected viruses had a median of six mosquito species and three mosquito genera (Fig 5). Moreover, geographical distribution per virus had a median of six countries. Moreover, all but one of the virus species (*Aedes vexans iflavirus*) were found in at least an additional mosquito species and an additional country (Fig 5). The two mesoniviruses detected in our study stood out in host range size with at least 22 mosquito species, more than twice the number of host species in any of the other viruses. Moreover, one of the mesoniviruses, *Alphamesonivirus 1*, had the widest geographical distribution by far, with detections in 18 countries and five continents. Other viruses were also widespread. For example, *Culex Iflavi-like virus 4* and *Culex pipiens-associated Tunisia virus* were found in more than ten countries and four continents.

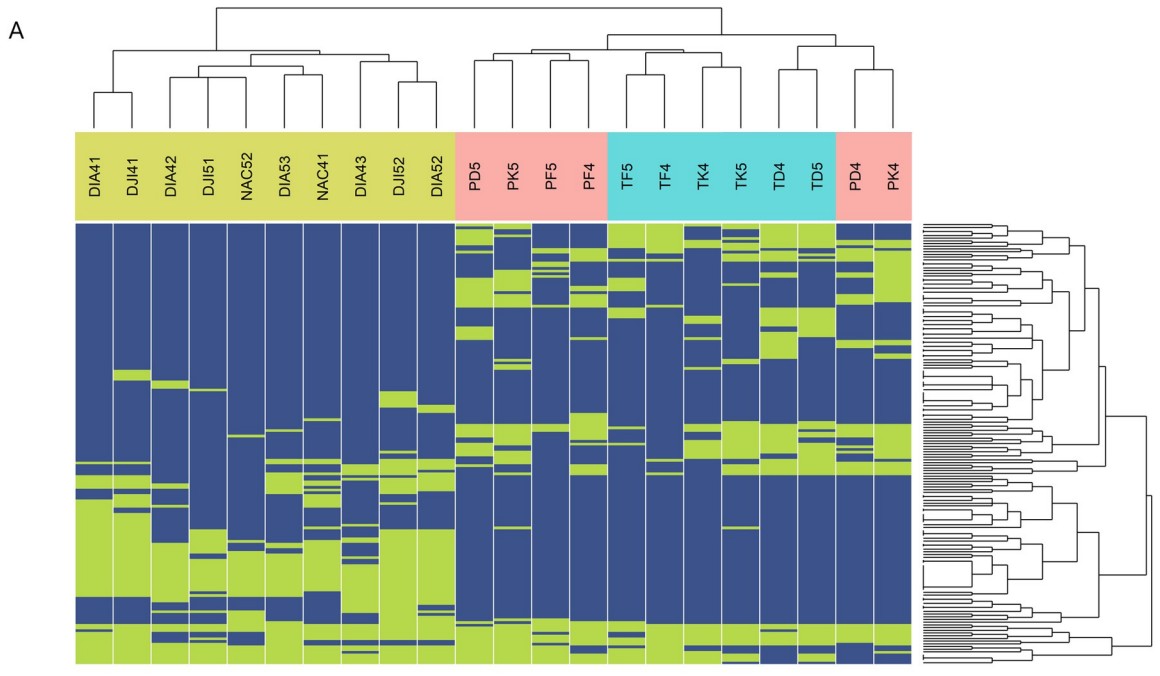

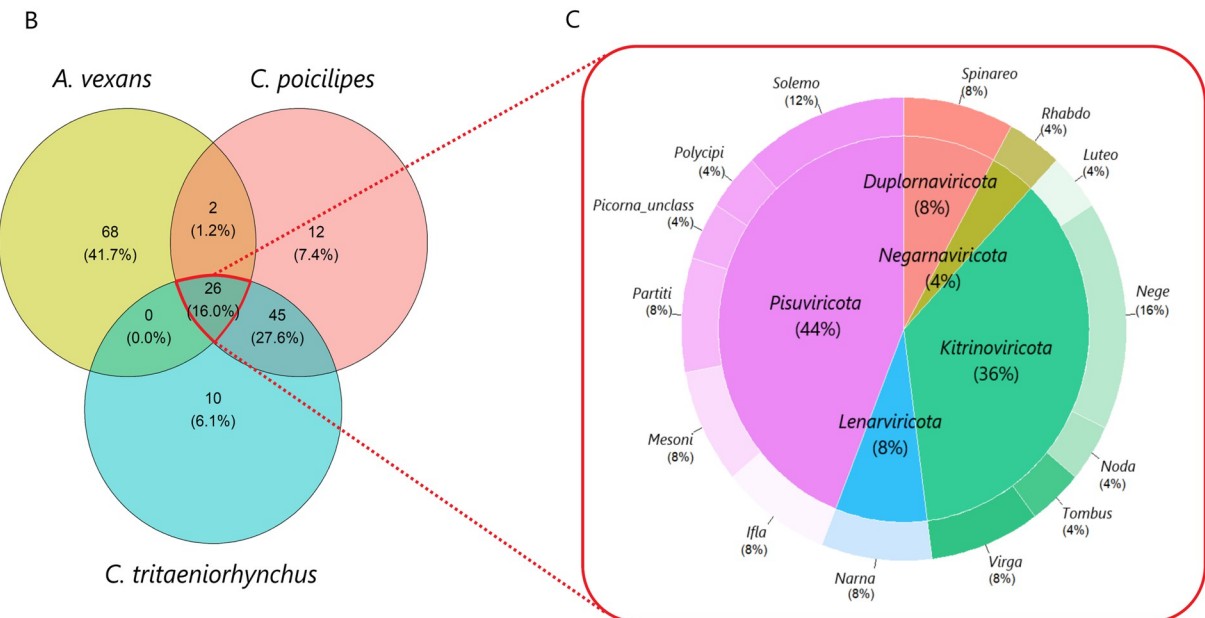

**Fig 4. Distribution of viral taxonomic units (VTUs) in three mosquito species. A.** Heatmap showing the presence (green) or absence (blue) of VTUs in libraries of the three mosquito species. Libraries and VTUs are ranked following a hierarchical clustering on the x-axis or the y-axis, respectively (dendrograms available on top and on the left of the heatmap). Library names are coloured following mosquito species (see Table 1 for explanation of acronyms), with libraries from *Aedes vexans* shown in green, *Culex poicilipes* in red and *Culex tritaeniorhynchus* in blue. **B.** Distribution of the 163 VTUs between the three mosquito species. Percentages between brackets represent the proportion of VTUs of each group in all VTUs. **C.** Distribution of the 26 VTUs shared by the three mosquito species (encircled in red) between clusters (outer donut graph) and phyla (inner pie chart).

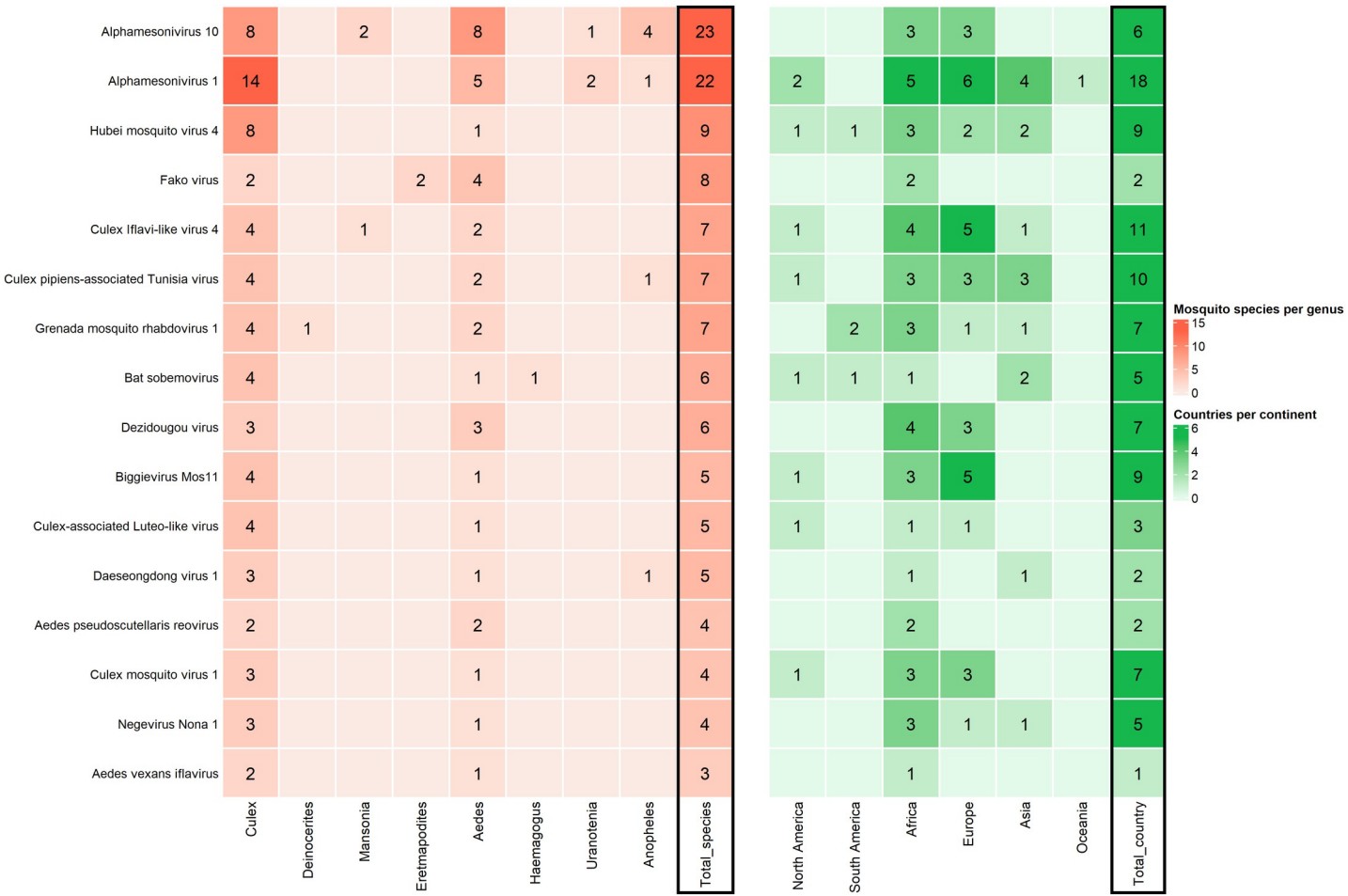

**Fig 5. Host range and geographical distribution of 16 known viruses belonging to the shared viral taxonomic units (VTUs).** Viruses are ranked on the y-axis according to the number of mosquito species in which they have been found (shown in the column "Total_species" on the right side of the left panel). **Left.** Number of mosquito species and genera associated to the viruses. Genera are shown on the x-axis below the heatmap. Numbers in tiles represent the number of mosquito species per genus in which a given virus has been detected. **Right.** Countries and continents in which the viruses have been detected. Numbers in tiles stand for the number of countries per continent in which a given virus has been detected. The total number of countries is shown in the last column on the right (Total_country).

## Discussion

### A new model for the influence of host taxonomy in the eukaryotic virome of mosquitoes

Our results show that the eukaryotic viromes of *C. poicilipes* and *C. tritaeniorhynchus* differed in the studied sites. Moreover, we did not detect a significant influence of the habitat on virome diversity and structure. These results were based on large mosquito collections in different and distant habitats along two sampling years. To our knowledge, one previous study has compared contemporary virome assemblies of sympatric mosquito species in different habitats [16]. Similarly to our results, Thongsripong and colleagues detected an influence of mosquito species, but not habitat, in the viromes of three mosquito species (*Armigeres subalbatus*, *Culex fuscocephala* and *Mansonia uniformis*) collected in sites situated at most 20 km apart over a year. Moreover, a few studies have compared the viromes of different mosquito species in several sites or regions, although without an explicit analysis of habitat influence [17,18,20,21]. These studies also point to an influence of the mosquito species on the eukaryotic virome. The

diversity of situations encompassed by those studies and ours strongly supports the generality of the influence of host taxonomy on the eukaryotic virome of mosquitoes.

The observed differences in virome diversity and structure could derive from three different scenarios. These scenarios are (i) differences in taxonomic diversity, (ii) differences in relative abundance of shared viruses, or (iii) a mixture of the two previous scenarios. Which scenario takes place is a question that is rarely explored in the mosquito virome. This lack of data is probably due to limits in experimental designs, especially in terms of numbers of mosquitoes and virus-like reads, that hamper in-depth virome characterizations. Here, intense efforts in mosquito sampling and sequencing depth allowed to identify 95 VTUs, belonging to 25 family-like clusters, in the *Culex* species. The experimental design also allowed to unveil that differences in relative abundance of shared taxa dominated in the viromes of the two *Culex* species. This observation supports the second scenario. Several factors could lead to these differences, like the mosquito or virus genomics or the mosquito microbiome. For example, differences in genomics between mosquito species can lead to differences in their immune systems, which in turn could lead to differences in the abundance of the same virus in the different hosts. It should be underlined that the observed virus abundances probably do not reflect true abundance. Our metatranscriptomics approach detects differences in relative abundance but, due to its limits as a quantitative method [23], additional methods with full quantitative power are required to robustly estimate abundance differences. Moreover, the clustering method can have led to errors in VTU demarcation (see Material and Methods). New clustering methods are required to avoid this problem and improve the identification of true virus species. Having said that, statistical analyses were done at the family level to avoid potential problems in VTU identification.

A dominance of differences in relative abundance of shared taxa has been previously observed between viromes of populations of the same mosquito species, *Culex pipiens*, depending on their geographical origin [31]. Moreover, several studies have shown that more than 40% of the virus species can be shared between the viromes of closely-related mosquito species [18,20,33,50]. Although differences in relative abundance were not explicitly analyzed in those studies, a high overlap in diversity suggests that such differences could both occur and dominate. On the other hand, the first scenario, that of a dominance of differences in virus diversity, is also supported by our diversity comparison between *Culex* and *Aedes* species and the literature [18,20]. For example, Shi et al. (2017) found that diversity overlap dropped from 60% to 13% with taxonomic distance when comparing the viromes of two *Culex* species and one *Aedes* species.

Taken together, these results depict a complex model of virome diversity and structure in mosquitoes based on host taxonomy. In this model, differences in relative abundance of shared viruses would dominate between the viromes of closely-related hosts (i.e. between populations of a mosquito species, or between closely-related mosquito species). The dominant difference would change along with host taxonomic distance. Differences in virus diversity would thus prevail between unrelated mosquito species. This preliminary model has yet limited empirical support. Given the interest of this first model for virome diversity and structure in mosquitoes, more studies exploring its validity in the studied mosquitoes and, more broadly, among the Culicidae diversity are required.

Despite the convergent results among studies on the influence of mosquito taxonomy on virome diversity and structure, our study presents limits that require additional work to fully validate our observations. More precisely, the number of libraries, although in the same order of magnitude found in many studies on the mosquito virome [16,18,19,49,50], was relatively small. This situation limits the robustness of the analyses of alpha and beta diversities. Moreover, our study has only explored a limited fraction of the habitat diversity exploited by *C.*

*poicilipes* and *C. tritaeniorhynchus*, and the habitats here were only replicated through sampling different years. Having said that, the large mosquito numbers warranted an in-depth exploration of virus diversity, a main goal of our study. It is necessary to develop experimental designs involving, among others, a larger number of samples and negative controls in order to further validate the results of this study.

## A large diversity of mosquito viruses have broad host ranges

We observed that more than 74% of VTUs were shared between the eukaryotic viromes of *C. poicilipes* and *C. tritaeniorhynchus* in any of the three habitats. The extent of diversity overlap was lower between the *Culex* species and *Aedes vexans* but still relatively consistent, with 17% of the 163 VTUs shared. This figure may not be fully exhaustive because we cannot rule out that other shared viruses were not detected. Nevertheless, the analysis of thousands of mosquitoes and the water scarcity favoring niche overlap between mosquito species should maximize the chances of identifying many of the viruses infecting the three mosquito species in the Ponds. Moreover, previous observations of shared viruses in other mosquito species strongly suggested that many of these viruses had a host range including several mosquito genera. The proportion of generalist viruses in our study was similar to those often found in virome comparisons between mosquito genera (*i.e.* 8–25% shared viruses in studies with replicate sites and more than 10 individuals per mosquito species) [15,18,20]. This consistent proportion of generalist viruses is remarkable given the differences in methodology, mosquito species and geographical regions among studies. These observations suggest that examples of generalism (i.e. a broad host range) might be common in mosquito viromes. Given the implications of this putative phenomenon on virus evolution and ecology, further exploration of the diversity of mosquito viromes is required to evaluate its generality.

No clear pattern was found in the distribution of generalist viruses along the taxonomy of RNA viruses infecting arthropods. In fact, the taxonomic diversity of those viruses was very high, including all five phyla of RNA viruses (*i.e.* Orthornavirae). There are several examples of a large taxonomic diversity among generalist viruses in the literature [17,20,51]. All together, these observations suggest evolutionary convergence in distant virus taxa towards host ranges including several mosquito genera. Moreover, the generalist viruses were often geographically widespread. For example, ten out of the 16 known viruses have been detected in at least three continents. The ubiquitous nature of certain virome members has been previously observed [18,50,51]. Two main scenarios, not mutually exclusive, can explain a quasi-global distribution: a global dispersion involving transmission between migrating mosquito species, or a co-radiation with the Culicidae along their evolution and spread [18,50]. Future phylogenetic analyses could shed light on the intriguing widespread distribution of generalist viruses in mosquitoes.

## Supporting information

**S1 Checklist. Inclusivity in global research questionnaire.**
(DOCX)

**S1 Fig. Rarefaction curve analysis of each library (see Table 1 for explanation of acronyms).** The x-axis shows the number of virus-like reads, and the y-axis the number of viral taxonomic units (VTUs) per library. **(A)** Libraries of *Culex poicilipes* (red) and *Culex tritaeniorhynchus* (blue). **(B)** Libraries of *Aedes vexans*.
(TIF)

**S2 Fig.** Distribution of viral taxonomic units (VTUs) among orders (external donut chart) and phyla (inner pie chart) found in (A) Culex poicilipes and Culex tritaeniorhynchus, and (B) Aedes vexans. Percentages between brackets represent the proportion of all VTUs in each order or phyla. The term "Incertae sedis" stands for taxa whose classification is still undefined at the phylum level.
(TIF)

**S3 Fig.** Average percent identities at the amino acid level from all the contigs of each viral taxonomic units (VTUs) with their best hit in the viromes of (A) Culex mosquitoes and (B) Aedes vexans. The bars in blue indicate an average percent identity higher than 90% and thus VTUs likely including sequences of the virus species found as their best hit. The red bars represent VTUs with less than 90% identity to their best hit and thus probably involving sequences of new virus species.
(TIF)

**S4 Fig. Distribution and relative abundance of viral taxonomic units (VTUs) in the libraries of *Culex poicilipes* and *Culex tritaeniorhynchus*.** Library names are indicated on top of the heatmap (see Table 1 for explanation of acronyms), along with a hierarchical clustering, and library colour indicates mosquito species (red: *Culex poicilipes*, blue: *Culex tritaeniorhynchus*). Tile colour stands for read abundance; the more abundant a cluster, the warmer the colour. The VTUs are ranked following total read abundance.
(TIF)

**S5 Fig.** Distribution of clusters (left) and viral taxonomic units (right) between mosquito species (A), sites (B) and years (C). Numbers between brackets stand for the proportion of each group among the total number of taxa.
(TIF)

**S6 Fig.** A. Distribution of cluster richness, Shannon and Simpson indices between libraries of Culex poicilipes (in red) and libraries of Culex tritaeniorhynchus (in blue). Dot color indicates the habitat while dot shape represents year. The significance of the comparison between distributions of the two species is shown above boxplots (Wilcoxon Mann-Whitney test). B. Non-metric multidimensional scale with Bray-Curtis dissimilarities obtained from the viromes of the two Culex species. Dot color indicates mosquito species and dot shape represents habitat. C. Comparison of Sorensen (for presence-absence data) and Bray-Curtis (for abundance data) dissimilarities between libraries. Each point therefore represents a dissimilarity index value between two libraries belonging each to a different mosquito species.
(TIF)

**S7 Fig.** Heatmap showing the presence (green) or absence (blue) of clusters in the different libraries. Libraries are ranked on the x axis following a hierarchical clustering (dendrogram available on top of the heatmap). Library names are coloured following mosquito species, with libraries from Aedes vexans shown in yellow, Culex poicilipes in red and Culex tritaeniorhynchus in blue (see Table 1 for explanation of acronyms). To facilitate visualization of shared clusters, the heatmap is separated into a top panel with the clusters only present in either the Aedes or the Culex species, and a bottom panel with the shared clusters.
(TIF)

**S8 Fig. Distribution of the highly-conserved viral taxonomic units (VTUs) over sites, mosquito species and years.** The VTUs are named after their best hit on the y axis. Each combination of mosquito species and site is presented on the x axis. Labels for the mosquito/site combinations are coded with the first letter standing for mosquito species (A for *Aedes vexans*,

P for *Culex poicilipes* and T for *Culex tritaeniorhynchus*), and the second letter for site (F for the Ferlo Region (Ponds), D for the Diama village (River) and K for the Keur Momar Sarr village (Lake)). Tile color stands for number of years with detection in a mosquito/site combination (red: Two years, yellow: One year, gray: No detection). The year of detection is provided within the tile whenever the virus was detected only one year.
(TIF)

**S1 Table. Literature analysis of the host range (Host) and geographic range (Country) of known generalist viruses found in this study.** The "Weighted_Avg_contigs_p_id" columns provide the average value of the identity percentages associated with the contigs. The column "Blastn" indicates other virus names given to a virus species in the NCBI database (e.g., different virus strains or naming errors during submission).
(XLSX)

**S2 Table. Output of the homology and taxonomy search for the Culex dataset.** The "Best-hit" column provides the accession with the lowest e-value found by Diamond. The "VTU" column contains the VTU names. The "cluster" to "genus" columns provide the VTU taxonomy as stated by the ICTV. All columns from "sum_reads" to "max coverage" provide information on the contigs of each VTU (column fields described below). *sum_reads* = total number of reads for each VTU, *Avg_match_length* = length of the alignment on the reference, *n_Contigs* = number of contigs associated with each VTU, *Avg_contigs_length* = average length of the contigs of each VTU, *Sum_contigs_length* = sum of all the contig lengths for each VTU, *Avg_Coverage_contigs* = average coverage of the alignment on the contigs, *Avg_Coverage_subject* = average coverage on the reference (alignment length / accession length), *Pond_Avg_contigs_p_id* = average percent identity at the amino-acid level of the contigs of a VTU, weighted by the length of the alignments, *min_contigs_p_id* = minimum percent identity at the amino-acid level of the contigs of a VTU with their best-hits, *max_contig_p_id* = maximum percent identity at the amino-acid level of the contigs of a VTU with their best-hits, *avg_contigs p_id* = average percent identity at the amino-acid level of the contigs with their best-hits, *min_coverage* = minimum read coverage, *max_coverage* = maximum read coverage, *Avg_read_depth* = average read depth for each VTU.
(XLSX)

**S3 Table. Output of the homology and taxonomy search for the Aedes dataset.** The "Best-hit" column provides the accession with the lowest e-value found by Diamond. The "VTU" column contains the VTU names. The "cluster" to "genus" columns provide the VTU taxonomy as stated by the ICTV. All columns from "sum_reads" to "max coverage" provide information on the contigs of each VTU (column fields described in the legend of S2 Table).
(XLSX)

## Acknowledgments

This work was funded by the European Union's Seventh Framework Programme through grant Vmerge (FP7-613996).

## Author Contributions

**Conceptualization:** Côme Morel, Patricia Gil, Geoffrey Gimonneau, Assane Gueye Fall, Biram Biteye, Marc Eloit, Serafin Gutierrez.

**Data curation:** Côme Morel, Biram Biteye.

**Formal analysis:** Côme Morel, Serafin Gutierrez.

**Funding acquisition:** Assane Gueye Fall, Momar Talla Seck, Marc Eloit, Serafin Gutierrez.

**Investigation:** Côme Morel, Patricia Gil, Baptiste Prepoint, Celine Condachou, Geoffrey Gimonneau, Biram Biteye.

**Methodology:** Côme Morel, Patricia Gil, Geoffrey Gimonneau, Biram Biteye.

**Project administration:** Patricia Gil, Assane Gueye Fall, Serafin Gutierrez.

**Software:** Antoni Exbrayat, Etienne Loire, Florian Charriat.

**Supervision:** Geoffrey Gimonneau, Assane Gueye Fall, Momar Talla Seck, Marc Eloit, Serafin Gutierrez.

**Writing – original draft:** Côme Morel.

**Writing – review & editing:** Côme Morel, Patricia Gil, Antoni Exbrayat, Etienne Loire, Florian Charriat, Baptiste Prepoint, Celine Condachou, Geoffrey Gimonneau, Assane Gueye Fall, Biram Biteye, Momar Talla Seck, Marc Eloit, Serafin Gutierrez.

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
