## [Decision Letter · Decision Letter 0]

7 Mar 2024

Host influence on the eukaryotic virome of sympatric mosquitoes and abundance of diverse viruses with a broad host range

PONE-D-23-37728

Dear Dr. Gutierrez,

We’re pleased to inform you that your manuscript has been judged scientifically suitable for publication and will be formally accepted for publication once it meets all outstanding technical requirements.

Kind regards,

Humberto Lanz-Mendoza

Academic Editor

PLOS ONE

Journal Requirements:

1. In your Methods section, please provide additional information regarding the permits you obtained for the work. Please ensure you have included the full name of the authority that approved the field site access and, if no permits were required, a brief statement explaining why.

"This work was funded by the European Union’s Seventh Framework Programme through grant Vmerge (FP7-613996)."

Please respond by return e-mail so that we can amend your financial disclosure and competing interests on your behalf.

3. We note that Figure 1 in your submission contain map/satellite image which may be copyrighted. All PLOS content is published under the Creative Commons Attribution License (CC BY 4.0), which means that the manuscript, images, and Supporting Information files will be freely available online, and any third party is permitted to access, download, copy, distribute, and use these materials in any way, even commercially, with proper attribution. For these reasons, we cannot publish previously copyrighted maps or satellite images created using proprietary data, such as Google software (Google Maps, Street View, and Earth). For more information, see our copyright guidelines: http://journals.plos.org/plosone/s/licenses-and-copyright.

Additional Editor Comments (optional):

Reviewers' comments:

Reviewer's Responses to Questions

**Comments to the Author**

1. Is the manuscript technically sound, and do the data support the conclusions?

Reviewer #1: Yes

Reviewer #2: Yes

2. Has the statistical analysis been performed appropriately and rigorously? 

Reviewer #1: Yes

Reviewer #2: Yes

3. Have the authors made all data underlying the findings in their manuscript fully available?

Reviewer #1: Yes

Reviewer #2: Yes

4. Is the manuscript presented in an intelligible fashion and written in standard English?

Reviewer #1: Yes

Reviewer #2: Yes

5. Review Comments to the Author

Reviewer #1: The authors present a viral metagenomic analysis of sympatric mosquito species collected over two years in three habitats in norther Senegal including habitats with permanent and temporary watercourses. Mosquitos, pooled by species (30 individuals aprox.), collection site and year, were used for RNA extraction and RNA was further pooled by these parameters, so that overall libraries were made from between 94 and 8640 individuals.

The results presented are clearly explained and technically adequate, though I can defer to a bioinformatic specialist on the suitability of the analysis pipeline, particularly with regards to the power to distinguish EVEs from infecting replicating viruses. This constitutes a general drawback in metagenomic analyses, namely that viral isolation is not attempted to confirm novel virus identification. Nevertheless, the results add important finding about the virome composition of mosquito species with little viromic characterization.

A drawback from the study is the lack of genomic and phylogenomic analyses. In addition, it is unclear whether the coverage of a viral genome was used in the criteria used to identify the presence of a VTU. Perhaps it is beyond the scope of the study, but placing viral sequences in their evolutionary context would enrich the study.

Statistical analysis was performed in an adequate manner using key metrics of virome diversity and structure. The species richness is a striking result from the analysis which contrasts with fewer viruses identified in different mosquito species, reported in the literature. This may be a result of the large number of individuals included in the libraries. Due to this factor in the study design, analysis of the variation in species richness and structure was not possible comparing smaller pools and these would have been important biological replicates. This limitation is discussed in the Discussion and considered in the study conclusions.

The data and previous reports support taxonomic rank as a key determinant of virome diversity and structure (relative abundance) as opposed to habitat. The methodological limitations of studying true abundance in different mosquito species, however, are discussed and additional data are required to support the proposed preliminary model of virome structure among related and distant mosquito species.

Reviewer #2: In this manuscript an analysis of the diversity of RNA viruses harbored by three mosquito species, comparing the sharing between them. For this study, shotgun sequencing for RNA viruses is conducted in a relevant number of individuals, looking for a correct sampling to identify the species of viruses, preliminary named as Virus Taxonomic Units and later identifying virus’s species comparing with data stored in the presently available databases. For the analysis was relevant the processing of data using statistical tools to normalize data, looking for a fair sampling, useful to propose a distribution model.

Comment:

In the paper Calle ML. Statistical Analysis of Metagenomics Data. Genomics Inform. 2019 Mar;17(1):e6. there is a criticism about the use of rarefaction as a normalization method. I suggest to include a comment in the discussion justifying why this method is used here.

6. PLOS authors have the option to publish the peer review history of their article (what does this mean?). If published, this will include your full peer review and any attached files.

Reviewer #1: No

Reviewer #2: No
